# RLIP76: A Structural and Functional Triumvirate

**DOI:** 10.3390/cancers13092206

**Published:** 2021-05-04

**Authors:** Jasmine Cornish, Darerca Owen, Helen R. Mott

**Affiliations:** Department of Biochemistry, University of Cambridge, 80 Tennis Court Road, Cambridge CB2 1GA, UK; jc2050@cam.ac.uk (J.C.); do202@cam.ac.uk (D.O.)

**Keywords:** RalBP1, RLIP76, small G protein, RhoGAP, Ral

## Abstract

**Simple Summary:**

The RLIP76 protein is present at high levels in multiple cancers, compared with levels in normal cells. In cancer cells it is thought to be important for removal of chemotherapeutics, while in normal cells it has been implicated in endocytosis, stress response and mitochondrial fission. Although RLIP76 is a potential target for ovary, breast, lung, colon, prostate and kidney cancers, only the middle third of the protein has been structurally characterized. Understanding the full structure of RLIP76 will help us to better understand the signalling pathways in which it is involved and by extension its role in cancer.

**Abstract:**

RLIP76/RalBP1 is an ATP-dependent transporter of glutathione conjugates, which is overexpressed in various human cancers, but its diverse functions in normal cells, which include endocytosis, stress response and mitochondrial dynamics, are still not fully understood. The protein can be divided into three distinct regions, each with its own structural properties. At the centre of the protein are two well-defined domains, a GTPase activating protein domain targeting Rho family small G proteins and a small coiled-coil that binds to the Ras family small GTPases RalA and RalB. In engaging with Rho and Ral proteins, RLIP76 bridges these two distinct G protein families. The N-terminal region is predicted to be disordered and is rich in basic amino acids, which may mediate membrane association, consistent with its role in transport. RLIP76 is an ATP-dependent transporter with ATP-binding sites within the N-terminus and the Ral binding domain. Furthermore, RLIP76 is subject to extensive phosphorylation, particularly in the N-terminal region. In contrast, the C-terminal region is thought to form an extensive coiled-coil that could mediate dimerization. Here, we review the structural features of RLIP76, including experimental data and computational predictions, and discuss the implications of its various post-translational modifications.

## 1. Introduction

RLIP76, also known as RalBP1, is a 76 kDa protein that was simultaneously discovered by three groups as a downstream effector of the Ral family of small GTPases. It was shown that RLIP76 binds to both RalA and RalB, but not to other closely related proteins within the Ras branch of the Ras superfamily of small GTPases [1,2,3]. Small G proteins cycle between a GDP bound, inactive state and a GTP bound, active state. They can slowly hydrolyse GTP, but for efficient hydrolysis a GTPase activating protein (GAP) is required to stimulate their intrinsic GTPase activity. A second set of proteins, the guanine nucleotide exchange factors (GEFs) stimulate the exchange of the bound GDP for GTP. The Ral proteins function downstream of Ras: Ras activates RalGEFs, which convert the Rals to their active, GTP-bound form. Ras proteins are mutated in approximately 19% of all cancers, with high incidences found in pancreatic (88%) and gastrointestinal cancers (colon, 50%; rectal, 50%; small intestine, 26%) [4], highlighting them as an important therapeutic target. Ras therapeutics however, have proven difficult to develop, although some progress is being made with covalent inhibitors against specific mutants (reviewed in [5]), and attention has turned to inhibiting downstream pathways, including the Ral pathway. The Ral proteins are drivers of cell growth, survival and metastasis [6]. Downstream of Ral, RLIP76 itself is overexpressed in multiple tumour types and mediates resistance to chemotherapeutics and apoptosis. It is itself a potential target in ovary, breast, lung, colon, prostate and kidney cancers (reviewed in [7]).

The RLIP76 protein can be divided into three distinct regions, each of which has its own interacting partners. Thus, it can be thought of as a protein of three parts, which may act separately or synergistically to perform the various functions that have been ascribed to RLIP76.

## 2. Domain Architecture of RLIP76

RLIP76 comprises 655 residues but contains only two recognizable domains in the centre of the protein (Figure 1). The RhoGAP homology domain encompasses residues 192–368 [1,8] and has been shown to stimulate the GTPase activity of the Rho family small GTPases Rac1 and Cdc42 in vitro [1,2,3], albeit very weakly. The Ral binding domain (RBD) was identified in a region adjacent to the RhoGAP domain and interacts with both RalA and RalB [2,9]. This was subsequently shown to form a short coiled-coil spanning residues 393 and 446, which is sufficient to bind tightly to the Ral proteins [9]. RLIP76 is not closely related to any other proteins, except within the RhoGAP domain, which is present in a large number of proteins with diverse domain arrangements [10]. The RhoGAP domain and Ral binding domain of RLIP76 are conserved within metazoans, whereas the N-terminal and C-terminal thirds are only conserved in *Danio rerio* through to *Homo sapiens* (Figure 1B) [11,12]. Ral binding is clearly an essential aspect of its function, since RLIP76 is only found in organisms that also have Ral [11,12,13].

The N-terminal third of RLIP76 (residues 1–190) binds to several other proteins. It interacts with the interdomain linker of the mu2 subunit of the clathrin adapter complex, AP2, leading to receptor mediated endocytosis of EGFR [14]. Residues 1–180 interact with the Arf GEF, ARNO, in turn activating Arf6, leading to cell spreading and migration [15,16]. Mutation of Ser29^RLIP76^ and Ser30^RLIP76^ to alanine inhibits ARNO binding [15]. A region in the N-terminal third of RLIP76 that extends 16 residues into the RhoGAP domain (1–208) has been shown to interact with the ENTH domain of the Epsin family of endocytic adaptors. Knockdown of Epsin has also been shown to decrease levels of active Arf6 and Rac1, since Epsin inhibits RLIP76 GAP activity [17]. Residues 1–392 of RLIP76, which includes the N-terminus and the RhoGAP domain, interact with the small GTPase R-Ras, leading to ARNO and subsequently Arf6 activation. The R-Ras binding site was subsequently narrowed down to residues 180–192, just N-terminal to the RhoGAP domain [16,18].

The C-terminal third of RLIP76, following the RBD, associates with REPS1 and POB1 (REPS2). REPS1 and POB1 are EH (Eps15 homology) domain-containing proteins that are involved in receptor-mediated endocytosis. The C-terminus of REPS1, residues 652–796, part of which (751–791) is predicted to form a coiled coil, interacts with the C-terminus of RLIP76 [19,20]. Similarly, the C-terminus of POB1 interacts with RLIP76. POB1 also binds to Epsin via its EH domain [21] and assembly of the RLIP76-POB1-Epsin complex leads to receptor mediated endocytosis via the clathrin adapter, AP2 [22]. During mitosis, endocytosis is switched off. It is proposed that RLIP76 acts as a scaffold that brings together the cyclin dependent kinase CDK1 and Epsin. CDK1 is then able to phosphorylate Epsin, preventing its interaction with AP2 and halting endocytosis [23,24]. Residues 497–655 of RLIP76 also associate with the mitotic cyclin, Cyclin B, which is found in association with CDK1 [23]. In addition, RLIP76 interacts directly with the large GTPase Drp1 (dynamin-related protein 1) through an undetermined region. Drp1 is required for mitochondrial fission during mitosis [25]. As with CDK1 and Epsin, RLIP76 acts as a scaffold and by forming a complex with both Cyclin B-CDK1 and Drp1, it facilitates phosphorylation of Drp1 by CDK1, leading to inhibition of mitochondrial fission. The inclusion of RalA in this complex targets and tethers RLIP76 to the mitochondrial membrane. Tight regulation of mitochondrial fission is required for the correct distribution of mitochondria during mitosis [25].

The C-terminus of RLIP76 also interacts heat shock factor 1 (HSF1), which regulates expression of heat shock genes in response to stress. The N-terminal DNA-binding domain of HSF1 is the minimal RLIP-binding region however inclusion of the HSF1 trimerization domain enhances binding. Activation of the Ral pathway leads to release of HSF1 from the regulatory complex formed with RLIP76 [26]. In postsynaptic neurons, the RLIP76 C-terminus interacts with the postsynaptic scaffold protein PSD-95, leading to endocytosis of PSD-95 associated receptors in synapses. Phosphorylation of rat RLIP76 at Thr645 (Thr653 in human RLIP76) inhibits PSD-95 binding [27].

## 3. The RBD and the Ral-RLIP76 Complex

The first insights into the structure of RLIP76 were the structures of the free RBD and the RBD-RalB complex (Figure 2). The Ral-binding domain forms a simple coiled-coil comprising two α-helices of approximately 20 amino acids each. Hydrophobic interactions between Leu, Ile and Val sidechains pin the coiled-coil together [9].

Small GTPases contain two switch regions that change conformation depending on the bound nucleotide. The RBD contacts RalB across both switch 1 and switch 2, with the switch 2 contacts being more extensive. C-terminal residues of switch 1 interact with helix 2 of the RBD, while switch 2 interacts with both helices of the RBD. Comparison of the free and bound RBD shows that, when bound to RalB, there are only small changes in the orientation of the helices and the loop between them. On binding RalB, helix 1 extends to include Gln417, resulting in the Gln being correctly orientated in the same direction as His413, which both then contact switch 2 of RalB (Figure 2A) [9].

RalA and RalB share 85% sequence identity and 100% identity in their RLIP76 binding regions. The energetic landscapes of the RBD binding interface on both RalA and RalB have been defined. Binding affinities of both RalA and RalB for RLIP76 RBD mutants showed that Leu409 and His413 in helix 1 and Trp430, Thr437 and Lys440 in helix 2 of the RDB are involved in binding to both RalA and RalB. However, some residues make different contributions to the complexes: Arg434 contributes to RalA binding but not RalB, while Leu429 contributes to RalB but not RalA binding. The differences likely arise from the dynamics of switch 2 of the Ral proteins, which is affected by sequence differences lying outside the identical region. One of the main contributions to the differences between RLIP76 binding to the Ral isoforms is a single amino acid insertion (Ala116) in RalB, although other non-conservative changes are present outside of the effector region [28].

The RBD has been used to develop peptidomimetics to inhibit Ral signalling and therefore downregulate one of the Ras effector pathways. Helix 2 was developed as a peptide inhibitor by introducing a hydrocarbon staple between unnatural amino acids introduced at residues 424 and 428 [29]. While this peptide bound to Ral proteins and inhibited RalB signalling in cells, the removal of the helix from the context of the coiled-coil led to a hydrophobic peptide that was prone to non-specific binding. Helix-stabilizing salt bridges were introduced to replace the hydrophobic residues that would normally make up the interface of the coiled coil, which improved its solubility and selectivity [29]. A selection was carried out to improve the RBD binding to Ral proteins, and mutants identified in this selection showed that the Gln433 to Leu substitution drove increased affinity (Figure 2B). The 20–30 fold affinity increase over wild type is due to both a contribution to the hydrophobic pocket around Trp430 and potentially the altered presentation of other RBD residues (Figure 2B). This study highlighted the significance of the hydrophobic pocket to the interaction with Ral proteins [29,30].

RalA is phosphorylated at Ser194 by Aurora A, which leads to its translocation to the mitochondria and increased interaction with RLIP76 [31]. Ser194 is not conserved in RalB. Although this implies that the C-terminus of RalA (177–206) might be involved in the RLIP76 interaction in vivo, full-length and C-terminally truncated RalA bind to the RLIP76 RBD with similar affinities in vitro [28]. It is more likely that Ser194 phosphorylation leads to co-localization of RalA and RLIP76, although it is possible that regions of RLIP76 outside of the RBD are also involved in the interaction and contact the RalA C-terminus, or that phosphorylation of Ser194 in RalA leads to differences to the dynamics of the switch regions [9,28].

Sites within the RLIP76 RBD that have been mutated in multiple patient samples are reported in cBioPortal and COSMIC, and include A396V, R414Q, D415H, R434I/S and L439P [32,33,34]. A396V, R414Q and D415H lie within the RBD but are unlikely to affect the Ral interactions since they are not in the binding interface. They could therefore be passenger mutations or could disrupt interactions with other regions of RLIP76. Arg434^RLIP76^ contacts Asp65^Ral^, which is an important interaction for RalA, but not RalB, binding. Mutation to either serine or isoleucine would disrupt this interaction [9,28]. A substitution of leucine to proline at position 439 would likely distort the helix 2 and cause reorientation of neighbouring Lys440. Mutation of this residue leads to reduction in RalA and RalB binding, so its reorientation could affect Ral binding.

## 4. The RhoGAP Domain

Structural information is also available for the RhoGAP domain, which was solved in tandem with the RBD and the 25-residue interdomain linker [8]. Biochemical evidence for RLIP76 GAP activity was observed at the same time as its discovery as a Ral effector [1,2]. Small GTPases are able to slowly hydrolyse GTP but require a catalytic arginine supplied by a GAP for efficient hydrolysis. This ‘arginine finger’ points towards the β and γ phosphates of GTP and stabilizes the transition state of hydrolysis. The RhoGAP domain of RLIP76 displays a characteristic fold of nine helices: A0, A, A1, and B-G and the arginine finger, Arg232, lies in the loop between helices A and A1 [8]. The primary arginine finger is stabilized by Lys268, which is within helix B and mutation of these two residues to alanine ablates GAP activity towards Cdc42 [35].

RLIP76 acts as a weak GAP towards Cdc42 and Rac1 in vitro compared to the canonical p50RhoGAP [8]. The overall fold and position of the arginine finger in RLIP76 and p50RhoGAP are similar but the RLIP76 RhoGAP domain binds to Cdc42 and Rac1 with affinities at least 100-fold lower. This lower affinity was thought to be due to loss of an important binding loop in the RLIP76 RhoGAP domain, but its reinstatement only slightly increased the binding affinity and did not increase the GAP activity. Structural data on the RLIP76 RhoGAP domain has therefore not provided the answer to its weak GAP activity. In cells the RLIP76 RhoGAP activity is more pronounced, as siRNA directed towards RLIP76 leads to increased levels of active Cdc42 and Rac1 [31]. RhoGAP domains display a broad spectrum of activity levels in vitro that do not always reflect their effect in vivo [10], presumably, because in vivo activity is mediated by more than just the isolated domains, with other regions of the proteins playing a role.

It is common for GAPs to contain other domains, for example PDZ domains, SH2 and SH3 domains that facilitate larger complex formation through further protein–protein interactions, and many RhoGAP proteins also contain specific lipid binding domains (PH, BAR, SAM) [10]. Membrane binding promotes the co-localization of the GAP and its cognate GTPases, as well as increasing their effective concentration, leading to more efficient hydrolysis [36]. Increased GAP activity in vitro has been observed for GAPs when either the GAP or GTPase is tethered to a membrane [37,38], so it is possible that RLIP76 has one or more membrane-localization motifs that could explain the higher GAP activity observed in vivo.

## 5. The RLIP76 RhoGAP-RBD Didomain

The RhoGAP domain and RBD exist in a fixed orientation to each other, held in place by a relatively inflexible linker. The nine residues (Gln369-Met377) following the canonical RhoGAP fold contact helices E and A and the loop between A0 and A in the GAP domain, on the opposite face to that involved in Rho protein engagement. Residues in the linker also interact with the RBD. Met377 contacts hydrophobic residues in the helix 1, whilst Trp382 and Met388 in the linker contact the helix 2. Together, these interactions serve to glue the GAP domain and RBD together [8].

The didomain structure led to a model showing dual engagement of RLIP76 with both Rho family proteins and Ral proteins [8,39]. Dual engagement relies on an appropriate orientation of both the Rho and Ral proteins, since both G proteins would be attached to the lipid bilayer via their C-terminal lipid modification and regions of the G-domain in contact with the membrane. Recent work on Ras has shown the existence of multiple orientation states of Ras at the membrane [40,41], some of which could be applicable to other small GTPases, although each GTPase may also have its own, unique membrane orientations. To accommodate dual engagement, Ral would need to adopt an orientation where the C-terminus of helices 3, 4 and 5 of the G domain contact the membrane. This orientation of Ral would have an exposed RLIP76 binding region, whilst keeping RLIP76 tucked close to the membrane. The Rho family protein could then contact the membrane via the C terminus of helix 3, the loop that follows, and helix 5 (Figure 3). More structural information on GTPases with their effectors and regulators at membranes will help to complete this picture. Tethering of RLIP76 to a membrane by Ral could increase its efficiency as a GAP by supporting correct localization, providing there is an orientation state that allows dual engagement [31]. This scenario would also support evidence that RhoGAP activity is stimulated upon RLIP76 binding to Ral·GTP in vivo even though it is not observed in vitro [1,8,31,42].

## 6. The N-Terminal Region

Outside the central domains, prediction software can give some indication of potential structure. The N-terminal third of RLIP76 (residues 1–192) is highly charged and proline rich. Visualizing this amino acid composition on a charge-hydropathy plot (PONDR), the RLIP76 N-terminus lies to the left of the cut off, indicating it is likely to be disordered (Figure 4A) [43,44]. IUPRED and PONDR scores by residue (Figure 4B) also indicate that the RLIP76 N-terminus is disordered [45,46,47,48,49]. Extending this analysis further, it is well established that many disordered proteins fold upon binding or contain short regions of structural propensity. These sites are termed molecular recognition features (MoRFs). ANCHOR predicts MoRFs based on whether there would be a gain in energy when a region binds an average globular protein, the parameters calculated are compared against a curated set of disordered proteins known to fold on binding and a score assigned from 0 to 1. For RLIP76, ANCHOR (IUPRED) predicts that the first 60 residues are likely to contain a MoRF, as well as other recognition regions from residues 100–120 and 150–164 (Figure 4B) [50,51]. Proteins that interact with the N-terminus, such as AP2, ARNO, R-Ras or ENTH, are candidates for species that could induce folding. In addition, post-translational modifications and membrane binding can also induce folding of disordered regions, which is discussed below [52,53].

## 7. The C-Terminal Region

For the C-terminal third of RLIP76, the COILS server predicts a coiled-coil between residues 500 and 600 and a shorter coiled-coil from residue 436–449, which corresponds to the second helix of the RBD (Figure 5) [54]. iTASSER structure prediction outputs a coiled coil, 208 amino acids in length across the whole C-terminal region [55,56], albeit with a low confidence score. Two of the top five models form a helix-turn-helix, while the remaining three form single coiled coils, across the same region. The RLIP76 binding partner POB1 utilizes resides 375–521 (short isoform numbering) to bind to the C-terminus of rat RLIP76, equivalent to human RLIP76 500–655 [21,22]. Residues 462–521 of POB1 also form a helix (as observed by CD), suggesting that the interaction between POB1 and RLIP76 could occur through an intermolecular coiled-coil [57]. Similarly, the binding region of HSF-1 overlaps with the binding region of POB1, so potentially also includes the RLIP76 coiled coil [26]. Interestingly, the trimerization motif of HSF-1 also forms a coiled-coil so this interaction may also involve formation of an intermolecular coiled-coil [58].

## 8. ATP Dependent Transporter Activity

The current structural data does not shed light on the ability of RLIP76 to act as an ATP-dependent transporter. Two ATP-binding regions were originally predicted, ^69^GKKKGK^74^ and ^418^GGIKDLSK^425^, based on their similarity to the ATP binding Walker motif GXXXXGK[TS] and the phosphoglycerate kinase P-loop sequence GGXKVXXK, respectively [59,60]. Mutation of Lys74 and Lys425 to methionine has been shown to abrogate ATPase activity and azido-ATP binding [60]. The first motif (69–74) lacks the threonine or serine present in Walker motif sequences, which chelates the essential magnesium ion [61]. The structural data on RLIP76 shows that the second ATP binding sequence (^418^GGIKDLSK^425^) is within the RBD, with Lys425 lying in helix 2 of the coiled-coil. Lys425 is not involved in Ral binding and would be available to contact the phosphate of ATP. We were unable to detect ATP binding to the RBD alone and suggest that additional contacts within RLIP76 would be necessary to contact the ribose and adenine base [9]. However, this would not be unusual; the Walker motif in ABC transporters requires a second ‘A loop’ motif, which consists of an aromatic amino acid (usually tyrosine) around 25 amino acids upstream of the Walker motif. RLIP76 has a tyrosine 27 amino acids upstream of the putative Walker motif [62]. More recent studies described an alternative potential ATP binding site on RLIP76. The site was identified by using docking to model ATP into a pocket coordinated by residues Gly235, Lys237, Ser265, Lys268 and Arg272, which are within the RhoGAP domain and roughly overlap with GTPase binding. However, this binding site has not been confirmed empirically [63]. Therefore, while ATPase activity or ATP-stimulated transport activity has been demonstrated in multiple studies [60,64,65], information outside of the central domains is needed to structurally characterize the ATP binding site in the N-terminus and to identify additional contacts that support Lys425 and underpin this ATP-binding site.

## 9. Localization

RLIP76 is present at the plasma membrane when localized by GTP-bound Ral but is also found throughout the cytosol and in the nucleus [66]. There is also evidence that RLIP76 has a cell surface epitope (residues 171–185), which was detected by live-cell confocal microscopy with anti-RLIP76 antibodies [67]. By sequence homology with a cell penetrating peptide, Penetratin, the region just N-terminal to this, 154–170 was suggested to be a potential transmembrane region [67]. This sequence is rich in lysine residues and ends with a polylysine stretch. As an isolated peptide, but not within the context of the whole N-terminal region, residues 154–170 are predicted to be cell penetrating by CPPred-RF, which is not surprising for such a highly charged sequence [68]. The de novo peptide structure predictor PEPFOLD3 predicts that the isolated peptide 154–170 folds into a short helix of 12 residues [69]. Together, these predictors describe a highly charged helix but not a region that would span the plasma membrane in the context of the whole protein.

Nevertheless, positively charged regions are often involved in contacting anionic phospholipids, and there is some evidence that membrane interaction can modulate the activity of other RhoGAP proteins. For example, DCL1 has a polybasic region that binds to PIP_2_ and increases its activity towards prenylated RhoA [38], while the CdGAP polybasic region binds PIP_3_, which enhances its GAP activity towards Rac1 [37]. Lipidation of the Rho protein is also an important component of the system, with the specificity of RhoGAPs in vitro changing depending on whether the GTPase is lipidated (and therefore membrane tethered) or not, and on the type of phospholipids present in the membrane [70]. Rac1 and Cdc42 expressed in insect cells (and therefore post-translationally modified) were stimulated by RLIP76 more than unmodified versions expressed in *E. coli* [42]. The RLIP76 region just N-terminal to the RhoGAP domain is the predicted α-helical peptide described above (154–170), which also includes a MoRF (150–164). This region is polybasic and includes six sequential Lys residues from 164–171. This polybasic region in RLIP76 could function similarly to that of DCL1 and CdGAP, mediating peripheral membrane association, rather than being permanently localized. Peripheral membrane association regions and transient helices that interact with anionic lipid heads are common features of disordered regions [71,72,73].

## 10. Quaternary Structure

Little is known about any quaternary structure formed by RLIP76, although there is some indication that RLIP76 may form a homodimer. Jullien-Flores et al. made a brief mention of unpublished data that indicated dimerization in yeast but this was never followed with published data [14] and a subsequent yeast-two hybrid study also found evidence of self-association [74]. We have not observed any evidence for dimerization via the RhoGAP or RBD [8], which leaves the possibility of dimerization through the N- and C-terminal sections. As the C-terminus is predicted to form a coiled-coil, this seems to be the most likely candidate for a dimerization motif. Disorder prediction software (PONDR/IUPRED) describes the C-terminus as disordered (Figure 4A) and indeed, at the sequence level, disordered regions and coiled-coils are similar in that they can both have low mean net hydrophobicity. Additionally, coiled-coils that dimerize are often disordered as a monomer [75]. Combining the predictions from IUPRED and COILS, it seems likely therefore that C-terminal the coiled-coil region could form a homodimer with another RLIP76 molecule.

## 11. Post-Translational Modifications

It was previously suggested that RLIP76 may be N-myristoylated, based on sequence inspection using PROSITE (Expasy) [76,77,78]. The predicted sites (Gly21, Gly40 and Gly191) however, being internal, would only be exposed to N-myristoyltransferase after proteolysis and are therefore unlikely to be biologically relevant. NMT-predictor and GPS-lipid are unable to predict N-terminal, or internal (post-cleavage) sites [79,80]. GPS-Lipid instead predicts a single palmitoylation site at the N-terminus of RLIP76 in the sequence ^1^MTECFLPPTS^10^ (Figure 6). Palmitoylation is a reversible lipid modification that can dynamically regulate protein membrane association and targeting [81,82]. This modification could allow for the cycling of RLIP76 between the cytoplasm and the plasma membrane.

RLIP76 is phosphorylated (Figure 6) in interphase, with further phosphorylation in the mitotic phase of the cell cycle [24]. Four PKCα phosphorylation sites (Ser118, Thr297, Ser353, Ser509) were identified by deletion mutant studies. Thr297 and Ser353 lie within the RhoGAP domain, whereas Ser118 and Ser509 are in the N- and C-terminal regions, respectively. Phosphorylation by PKCα led to an increase in Doxorubicin-transport activity and this effect was partially abrogated by deletion of residues 297–299. This region is however within the RhoGAP domain, so it is also possible that the activity was affected by improper folding or removal of GAP activity [83]. Thr297 is on the opposite face of the RhoGAP domain to Rho protein binding, so would be accessible to a kinase even when in complex with a Rho family protein. Ser353 is the final residue in the loop that is truncated in the RLIP76 GAP domain compared to p50RhoGAP, so may have a function in binding to Rho family proteins. RLIP76 specific mass spectrometry analysis revealed 14 phosphorylation sites in total, including Ser118 [84]. The same 14 sites have also been observed in high-throughput mass-spectrometry studies (Table 1). In addition to Thr297 and Ser353, phosphorylation of Ser252/Thr253 (which could not be discriminated) within the RhoGAP domain was also detected. Ser252 points away from the core of the protein, so if phosphorylated it would not affect the GAP domain structure. In contrast, Thr253 forms a hydrogen bond with Glu215, which would be disrupted by phosphorylation, implying some change in the structure. Phosphorylation of other RhoGAP domains can cause inhibition or stimulation of GAP activity, including dimerization-dependent increased GAP activity, or can change the specificity from one Rho family protein to another [85,86,87], so it is possible that one or more of these observed modifications will affect the activity of RLIP76.

In the C-terminus, phosphorylation of Ser463 and Ser645 of RLIP76 was shown to occur in response to RalB overexpression [84]. Significant changes in phosphorylation of other residues, including those in the RhoGAP domain, were not detected, suggesting that their modification may occur in response to different stimuli. Some of the sites were found phosphorylated together in the same peptide fragment, indicating that phosphorylation of one site may depend on another or, alternatively, some sites are constitutively phosphorylated while others are regulatable [84]. Phosphorylation of rat RLIP76 Thr645 by PKA (equivalent to Thr653 in the human protein) inhibits PSD-95 binding and the phosphomimetic mutant (T653E) of human RLIP76 also fails to form a complex with PSD-95. Dephosphorylation of RLIP76 occurs in response to NMDA receptor activation, allowing association of RLIP76 with PSD-95 and internalization of AMPA receptors [27].

Intriguingly, over half of the annotated phosphorylation sites are in the N-terminus of RLIP76 (Figure 6). In addition to changing signalling pathways, as with PSD-95, phosphorylation can also affect protein structure, especially of disordered regions. Phosphorylating disordered regions can affect their backbone dynamics and cause a shift towards one particular state of a population of disordered structures, or even cause folding [52,73]. It seems therefore likely that these phosphorylation sites function to induce some structural rearrangement.

## 12. Conclusions

Despite its discovery over 25 years ago, the amount of experimental structural data available for RLIP76 is still limited to the middle third of the protein. This is likely due to the flexible nature of the full-length protein that precludes high resolution X-ray analysis but is exacerbated by the paucity of recognizable domains. The information that we have about the structural features of RLIP76 along with the predictions based on the sequence information and potential post-translational modifications are summarized in Figure 6. Experimental information about the N- and C-terminal thirds of the protein will be essential for understanding their role in interactions with other molecules, the effects of their post-translational modification and their roles in the multiple functions of RLIP76.

## Figures and Tables

**Figure 1 cancers-13-02206-f001:**
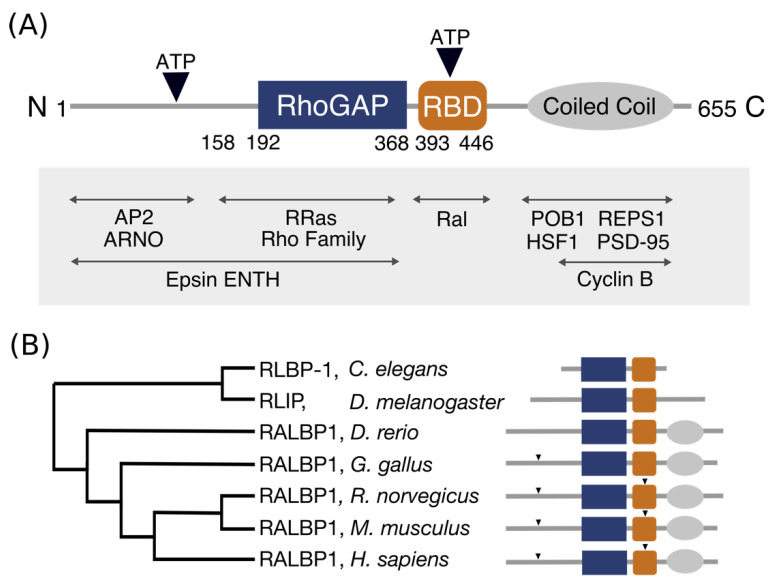
Architecture and conservation of the domains in RLIP76. (**A**) Domain architecture of RLIP76. The central RhoGAP and Ral binding domains are flanked by structurally uninvestigated N- and C-termini. The C-terminus is predicted to form a coiled coil, while the N- terminus is predicted to be disordered. The positions of two ATP binding sites are indicated by inverted triangles. Known interactions are labelled, with their approximate binding regions indicated by arrows. (**B**) Phylogenetic tree of RLIP76 in a subset of organisms. The schematic (styled as in Figure 1A) shows the conserved features for each organism. Only the RhoGAP and RBD are conserved in *Caenorhabditis elegans*. The predicted coiled-coil is present in *Danio rerio* through to *Homo sapiens*, while the two ATP binding motifs are present in *Rattus norvegicus*, *Mus musculus* and *Homo sapiens*.

**Figure 2 cancers-13-02206-f002:**
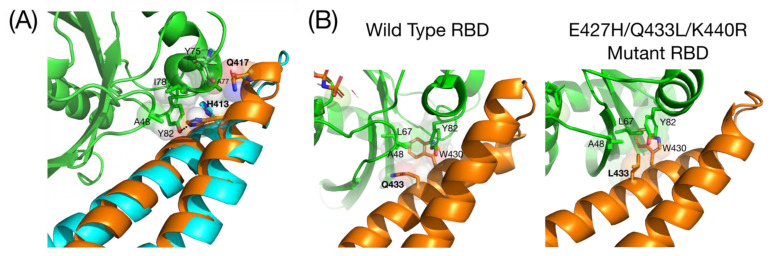
Structures of the free RBD and the RBD-RalB complex. (**A**) Comparison of the free RBD (cyan) superimposed on the RBD RalB complex (RBD = orange, RalB = green) (2KWH, 2KWI). The helix in the bound RBD extends, with Gln417 pointing in the same direction as His413. Interactions involving His413 and Gln417 are shown with the sidechains in spacefilling representation superimposed with sticks: carbons are the same colour as the ribbon, oxygen is red and nitrogen is blue; (**B**) Comparison of the hydrophobic pocket of WT RBD and the tighter binding E427H/Q433L/K440R mutant (6ZQT). Residues in the hydrophobic pocket round Trp430 is shown with the sidechains in spacefilling representation superimposed with sticks. Colour scheme is the same as in (**A**).

**Figure 3 cancers-13-02206-f003:**
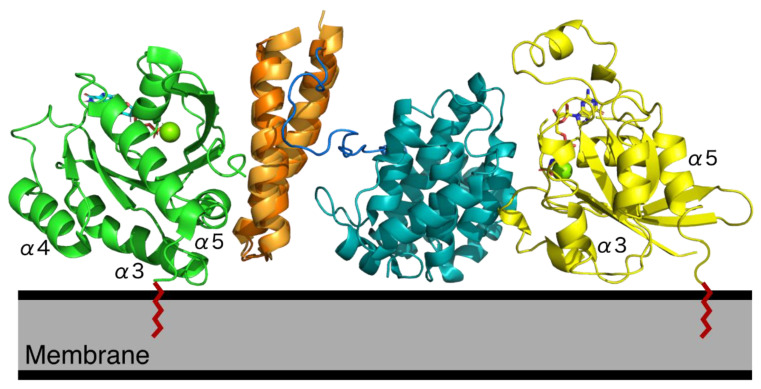
RLIP76 RhoGAP and RBD interactions. RLIP76 could interact simultaneously with both Ral and Rho family proteins. Cdc42 was docked to the RLIP76 didomain (2MBG) using the Cdc42-p50RhoGAP complex to overlay the RhoGAP domains (1AM4). The RBD of RLIP76 of this model was then superimposed with the RBD in the RalB complex (2KWI). The RLIP76 RBD is orange, the RhoGAP domain is cyan, the RLIP76 linker is blue, RalB is green and Cdc42 is yellow. Helices in RalB and Cdc42 that would contact the membrane in this orientation are labelled and their C-terminal lipid modifications are shown in red.

**Figure 4 cancers-13-02206-f004:**
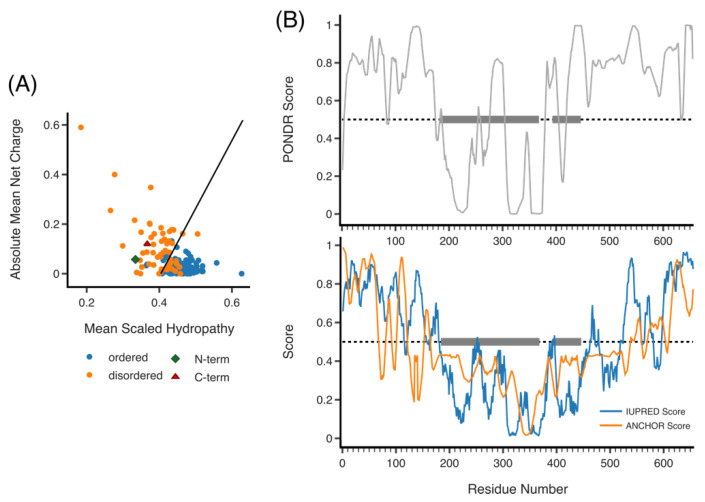
Predicted disorder of RLIP76. (**A**) Hydropathy plot of RLIP76 N- and C-termini (green diamond and red triangle, respectively), the remaining plotted proteins (ordered, blue and disordered, orange) are from the PONDR data set. RLIP76 lies to the left of the boundary line for disordered proteins. (**B**) PONDR plot of disorder across residues 1–655 (full length) RLIP76 (top). IUPRED plot of disorder (blue) and ANCHOR score (orange) across residues 1–655 (bottom). In both PONDR and IUPRED scores above 0.5 are disordered. ANCHOR scores above 0.5 predict regions that are likely to fold upon binding. Grey boxes highlight the locations of the RhoGAP and RBD.

**Figure 5 cancers-13-02206-f005:**
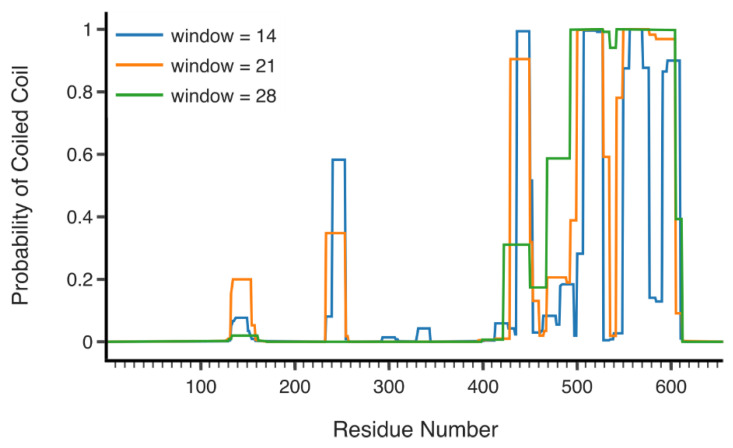
COILS prediction of RLIP76. Plot of predicted coil coiled regions in RLIP76 by COILS server. Window size is the number of amino acids in the sliding window the server uses across the sequence. The resolution between globular and coiled-coil score distributions decreases strongly with decreasing window size. However, ends of coiled coils and short sections can be identified with smaller windows.

**Figure 6 cancers-13-02206-f006:**
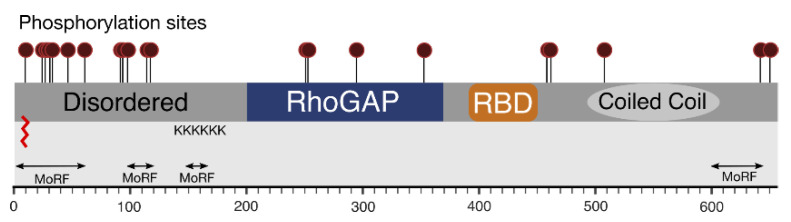
Compiled knowledge of the structure of RLIP76. Phosphorylation sites are shown as maroon lollipops. Predicted and known regions of structure are boxed. The predicted palmitoylation site is shown as a red zig-zag. The poly-lysine tract and predicted molecular recognition features are also shown.

**Table 1 cancers-13-02206-t001:** Phosphorylation sites in RLIP76. High-throughput studies with five or more references on PhosphoSite and those that were found through RLIP76 specific studies (LTP) are shown.

Site	Location in Sequence	Number of References: LTP/HTP
**SER11**	N-terminus	1/9
**THR27**	N-terminus	1/49
**SER29**	N-terminus	1/538
**SER30**	N-terminus	1/103
**SER34**	N-terminus	1/76
**SER48**	N-terminus	1/57
**SER62**	N-terminus	1/59
**SER92**	N-terminus	1/41
**SER93**	N-terminus	1/45
**SER99**	N-terminus	1/5
**SER116 ^1^**	N-terminus	1/9
**SER118 ^1^**	N-terminus	2/3
**SER252 ^1^**	RhoGAP	1/0
**THR253 ^1^**	RhoGAP	1/0
**THR297**	RhoGAP	1/0
**SER353**	RhoGAP	1/0
**SER461**	C-terminus	0/10
**SER463**	C-terminus	1/36
**SER509**	C-terminus	1/0
**SER645**	C-terminus	1/14
**THR653 ^2^**	C-terminus	1/0

^1^ Ser116/Ser118 and Ser252/Thr253 sites were too close to determine which was phosphorylated [84]. ^2^ Found as phosphorylation of rat RLIP76 at Thr645.

## Data Availability

Not applicable.

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
