# Peer review of "RLIP76: A Structural and Functional Triumvirate"

_cancers, 2021, doi:10.3390/cancers13092206_

Round 1
Reviewer 1 Report
This is a very well written review describing the structural and functional properties of RLIP76 including the details about RLI76 interacting proteins.
I have the comment about the ATP-binding domains of RLIP76 which the authors should address since ATPase activity is one of the important and most explored function of RLIP76 (Figure 1).
In Figure 1, ATP-binding domains have not been shown and the location of the ATP binding site is shown as yet to be confirmed. It has been shown by Awasthi et al. (Functional reassembly of ATP-dependent xenobiotic transport by the N- and C-terminal domains of RLIP76 and identification of ATP binding sequences. Biochemistry. 2001 Apr 3;40(13):4159-68. doi: 10.1021/bi002182f. PMID: 11300797), That RLIP76 has two distinct ATP-binding domains. The ATP binding sites in N-RLIP76(1-367) and C-RLIP76(410-655) were identified to be (69)GKKKGK(74) and (418)GGIKDLSK(425), respectively and mutation of K(74) and K(425) to M residues, in N-RLIP76(1-367) and C-RLIP76(410-655), respectively, abrogated their ATPase activity as well as azido-ATP labeling. The ATPase activity, or ATP-stimulated transport activity has been confirmed by mutation studies in N-RLIP76(1-367) and C-RLIP76(410-655) which lose the activity individually and retail the ATPase activity by mixing both N-RLIP76 and C-RLIP76. The transport functions of RLIP76 were shown and confirmed in several papers including recent paper by Sharad S Singhal et al. Carcinogenesis, 2021;, bgab016, https://doi.org/10.1093/carcin/bgab016.
Alternate binding sites based on docking to model ATP into a pocket coordinated by residues Gly235, Lys237, Ser265, Lys268 and Arg272 are shown by Awasthi et al. (Oncotarget. 2018; 9: 36202-36219) but were not confirmed by their mutation studies as mutants M1, M3 and M4, deletion mutations were not on the potential ATP-binding sited based on their docking studies (Table 1, Oncotarget. 2018; 9: 36202-36219).
M1 (Y231A;R232A;V233K,S234A)
M2 (A264R;S265A;L267A;K268A)
M3 (Q340A;N341A)
M4 (I344A;V345A)
Overall this is very well written review describing structural ad functional properties of RLIP76 and its interaction with other proteins.
Author Response
We are grateful to the reviewer for their positive comments.
Thank you for your comments about the lack of clarity on ATP binding. The omission of the binding sites from Figure 1 was an oversight that has now been corrected. We have also added another sentence into the text, to highlight the mutations of K74 and K425 and their effects on the ATPase activity. Furthermore, we have clarified that the computationally determined ATP binding site within the RhoGAP domain has not been confirmed empirically. At the end of the paragraph on ATP binding we have reworded the final sentence, to explain that although the ATP binding sites have been partially elucidated, there is not structural data yet on ATP binding.
Reviewer 2 Report
- the disease significance of RLIP76 is not presented well enough
- the abstract should be rewritten systematically.
- The authors have not shared how this protein in cancer biology
- What are the other family members of this protein in other species
- A Phylogenetic tree should be presented
Author Response
1. the disease significance of RLIP76 is not presented well enough
We agree that if this were a review on RLIP76 in general, the disease significance of the protein would be a much greater part of the manuscript. However, we were invited to write a review specifically on the structural aspects of the protein, which would form part of a special issue on p53 and RalBP1/RLIP76 in carcinogenesis. We therefore expected that other review articles within the special issue would cover the disease significance of RLIP76 extensively. To avoid repetition of the same material within several articles in the same special issue we have confined our review to the structural aspects, as requested.
2. the abstract should be rewritten systematically.
We apologise if the abstract was not clear enough. We are not sure what changes are requested by the reviewer, so we have modified the abstract to summarise the material in the review more explicitly.
3. The authors have not shared how this protein in cancer biology
This point overlaps with point 1, and we have the same response: that the role of RLIP76 in cancer biology will be covered extensively elsewhere in the special issue.
4. What are the other family members of this protein in other species
5. A Phylogenetic tree should be presented
RLIP76 is the only member of its family, based on the sequence of the protein. There are of course, several other RhoGAP proteins, which each have their own unique domain structures. We have stated this at the end of the paragraph in the section entitled ‘Domain architecture of RLIP76’ and added a reference to a recent paper that includes these domain structures for the interested reader. We have also included information about the orthologues of RLIP76, which show that the central domains are conserved throughout metazoa, with worms and fruit flies having shorter N- and C-termini. Both ATP binding motifs and the N- and C-terminal extensions are only present in mammals. This information is summarised in a simplified phylogenetic tree, along with a schematic of these features, in Figure 1B.
Round 2
Reviewer 1 Report
Authors have revised the manuscript thoroughly.